# Genesis of the Baiyun Gold Deposit in Northeast Hubei Province, China: Insights from In Situ Trace Elements and S-Fe Isotopes of Sulfide

Weifang Song [1,2], Jianzhong Liu [3,4,5,*], Yuanbing Zou [1,2,*], Xingping Liu [1,2], Taocheng Long [1,2], Jiandong Zhu [1,2], Shengbo Fu [1,2], Song Chen [1,2], Yangfu Xiong [1,2], Runjie Zhou [2,6], Jingjing You [1,2], Xinqi Zhou [1,2], Zaixi Yang [1,2] and Jie Fang [1,2]

[1]  NO. Six Geological Team of Hubei Geological Bureau, Xiaogan 432000, China; gs.wfsong19@gzu.edu.cn (W.S.); dragontc@163.com (T.L.); 18963975955@163.com (J.Z.); 13669029269@163.com (S.F.); fengkuang2352006@126.com (S.C.); xf_139@163.com (Y.X.); youjing19921110@163.com (J.Y.); xinqizhou2024@163.com (X.Z.); chinayangzaixi@163.com (Z.Y.); 13477623737@163.com (J.F.)

[2]  Hubei Key Laboratory of Resources and Eco-Environment Geology, Xiaogan 432000, China; rjzhou@foxmail.com

[3]  Guizhou Bureau of Geology and Mineral Exploration & Development, Guiyang 550004, China

[4]  Innovation Center Ore Resources Exploration Technology in the Region of Bedrock, Ministry of Natural Resources of the People's Republic of China, Guiyang 550018, China

[5]  College of Resource and Environmental Engineering, Guizhou University, Guiyang 550025, China

[6]  Hubei Geological Survey, Wuhan 430034, China

*  Correspondence: m13765115603@163.com (J.L.); zouyuanbing@163.com (Y.Z.)

**Abstract:** The Baiyun gold deposit is a medium-sized deposit in northeastern Hubei around the southern margin of the Tongbai-Dabie metallogenic belt. However, its genesis has not been determined. The metallogenic process of the Baiyun gold deposit can be divided into three stages: quartz + feldspar, quartz + native gold + electrum + polymetallic sulfides, and quartz + pyrite + calcite + iron dolomite + illite. In this study, LA-ICP-MS was used for in situ trace element and isotope analyses in the main and late ore stage hydrothermal sulfides to evaluate the genesis and evolution of ore-forming fluids. Gold is positively correlated with Ag, Cu, Pb, Zn, and Te and the Co/Ni ratio is greater than 1. The S isotope values of Py1 and Py2 are −0.23–3.04‰ and 1.27–6.09‰, respectively. As mineralization progressed, S isotope values increased. In situ S isotope values of the two types of galena symbiotic with pyrite in the main metallogenic stage are 2.97–3.47‰. In situ Fe isotopic values of pyrite are −0.05–0.82‰; values in the two stages are similar without significant fractionation. We inferred that the Baiyun gold deposit formed via magmatic mineralization related to the subduction of the Pacific Plate during the Yanshanian.

**Keywords:** pyrite; galena; in situ analysis; trace elements; S-Fe isotopes; Baiyun gold deposit

## 1. Introduction

The Tongbai-Dabe metallogenic belt is an important site of precious metals, rare metals, and strategic non-metals in China [1–9], and the Baiyun gold deposit in northeastern Hubei Province is the largest gold deposit in the southern margin of the metallogenic belt. This deposit has important typicality and representative characteristics for understanding gold mineralization in the area [10–13]. Previous studies of the Baiyun gold deposit have mainly focused on the geological characteristics [10–13], trace elements in the ore and surrounding rock [10,11], occurrence of gold [10–13], alteration of the surrounding rock [10–13], sulfur isotopes of sulfides [10–13], and fluid inclusions and hydrogen and oxygen isotopes [10]. It has been preliminarily identified as a magmatic-hydrothermal deposit [10–13]. However, data related to the genesis of the deposit are limited. Furthermore, previous research was

mainly conducted at the end of the last century, with few recent studies of this deposit. Thus, advanced in situ analysis technology and more accurate geochemical data are required.

Pyrite and galena are important hydrothermal sulfides in the paragenetic sequence with the main ore minerals (natural gold and electrum). Therefore, the study of these sulfides is of great significance for understanding the geochemical characteristics of deposits. However, the structural characteristics, trace elements, and isotopic compositions of sulfides differ significantly with respect to environmental conditions. Previous studies have suggested that in situ trace elements and isotopic compositions of hydrothermal sulfides are important for determining the source of ore-forming fluids in medium- to low-temperature hydrothermal gold deposits [14–22]. Moreover, the surrounding rock of the Baiyun gold deposit is Neoproterozoic granodiorite, Early Cretaceous lamprophyre, Huangmailing Formation strata, and Qijiaoshan Formation strata of the Neoproterozoic Hongan Rock Group and the stratigraphic lithology is mainly dolomite albitite gneiss, quartz schist, albitite hornblende schist, albitite granulite, light granulite, manganese marble, graphite schist, and epidotite albitite hornblende schist, and part of the surrounding rock is of magmatic origin [10–13]. Therefore, sulfur isotope values of sulfides may be similar to those of magmatic sulfur, indicating that the S isotopes of sulfide powder cannot provide reliable data for understanding the genesis of the Baiyun gold deposit. In this study, in situ trace element analyses and isotope testing of the hydrothermal sulfides in ores by LA-(MC)-ICP-MS (laser ablation multi collector *inductively* coupled plasma *mass spectrometry*) were performed to reveal the source and evolution of ore-forming fluids. Moreover, characteristics of geochemical data were analyzed to further discuss the genesis of ore deposits from the perspective of microzones.

## 2. Geological Background

### *2.1. Regional Geology*

2.1.1. Tongbai-Dabie Metallogenic Belt

The Tongbai-Dabie metallogenic belt is located between the North China Block and the Yangtze Block. It represents the eastern extension of the Qinling composite orogenic belt (an important part of the Central Orogenic Belt of China) and is one of 26 important metallogenic belts in China [23–31]. It belongs to the southern Qinling-Huaiyang fold belt of the Qinling fold system and can be divided into three secondary tectonic units: southern Qinling, Tongbai-Dabie, and northern Huaiyang metallogenic belts [24,30]. The Precambrian metamorphic rock series is largely exposed in the belt, and magmatic activity is active, with long invasion and eruption times. Magmatic rocks of various types are widely distributed in the belt. Intermediate-acid intrusive rocks are mainly exposed in the Mesozoic, followed by basic–ultrabasic rocks, while neutral rocks are rare [13]. Medium-low metamorphism occurred in the Wudang-Suizhou and northern Huaiyang areas and medium-high metamorphism occurred in the Tongbai-Dabie area. The Dabie area is one of the largest and best-preserved ultrahigh-pressure metamorphic bodies in the world [23,26,27,29]. The northern margin of the metallogenic belt of Henan Province hosts the Qianechong, Doupo, and other large molybdenum deposits and the Laowan and Bodaoling large-super to large gold and silver deposits [6,32–35]. The North Huaiyang metallogenic belt of Anhui Province hosts the Shapinggou and other super large molybdenum deposits [1,5,7,9,36]. The southern margin of the metallogenic belt in northeastern Hubei Province hosts the Baiyun gold deposit, Niangniangding beryllium deposit, Wangjiawuji tungsten deposit, Chenlingou gold deposit, and Huahe large fluorite deposit [10–13,37] (Figure 1). The output of these deposits is closely related to the Yanshan intrusion in time and space.

Our research group and others have studied the typical deposits produced by the above intrusions and their peripheries extensively, revealing several key results. (1) Highly differentiated granitic intrusions related to regional rare metal mineralization were mostly formed in the Yanshan period [38,39]. (2) The main genetic types of rare metal deposits produced in the study area include porphyry, altered rock, and quartz veins. There is

general support for a magmatic-hydrothermal genesis with mineralization in the late Yanshan period; the mineralizing rock mass is a highly differentiated intermediate-acid granitic rock mass [40–50]. (3) In terms of metallogenic series, the study area mainly shows an Early Cretaceous Mo-Be-W ore-forming subseries, an Early Cretaceous Au-Ag ore-forming subseries, and a late Early Cretaceous Mo-Pb-Zn ore-forming subseries. The above metallogenic subseries belong to a large metallogenic series and are related to Pacific Plate subduction and retreat in metallogenic dynamics in the Yanshan period [1–5,51–53]. (4) Some researchers believe that the material source of molybdenum deposits in the Dabie region was the mantle, while the metallogenic fluid was sourced from the lower crust, rather than from the fluid metallogenic differentiation of a large rock base and rock mass produced near the surface. The contact belt of the rock base, rock mass, and veins exposed on the surface only provided a channel and space for mineralization [54]. (5) From the rock base to the inner and outer contact belts and then to the distal end, the evolution trend was from high-temperature deposits (Mo, Be, W, and Li) to medium-temperature (Au and Ag) and low-temperature (fluorite and barite) deposits, which were the product of the same metallogenic system in different locations. The northern and southern margins of the Tongbai-Dabie metallogenic belt had similar metallogenic tectonic backgrounds. However, the types of deposits produced are different, and there are differences in the ore-controlling factors, mineralization forms, and metallogenic ages. Among these, gold, molybdenum, tungsten, beryllium, fluorite, and other strategic minerals as the dominant minerals in the metallogenic belt have been the focus of research.

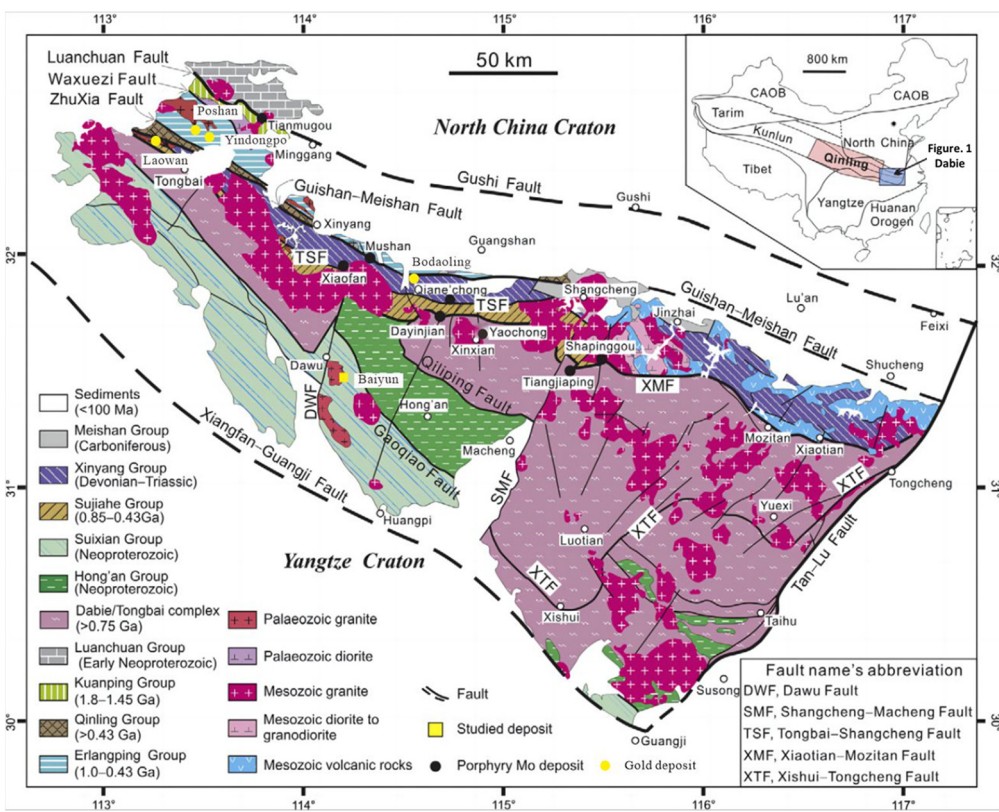

**Figure 1.** Diagram of geology and deposits in the Tongbai-Dabei metallogenic belt (modified from [2]).

The Baiyun gold deposit in Dawu County is accompanied by silver mineralization and is representative of the area. The Baiyun gold deposit is controlled by the NW and NE fault fracture zones and contact zones between different rock masses (veins), the orebodies extend to a deep level without a sealing edge, and the depth still has prospecting potential. Gold and silver orebodies have been found in the secondary structures parallel to the main ore-bearing structures in the NW and NE directions of the Baiyun gold deposit, indicating

that the gold mineralization intensity was high and the geological conditions for gold mineralization were good in the area [10–13]. Partial studies on trace elements in ores and surrounding rocks, sulfur isotopes of sulfide powder and fluid inclusions in quartz veins have yielded results. For example, the formation of the Baiyun gold deposit may be related to granitic magmatic activity during the Yanshanian [10,11]. However, systematic and reliable geochemical data are lacking, limiting analyses of the source of ore-forming materials, the genesis of the deposits, and the potential for deep-edge prospecting in mining areas.

### 2.1.2. Baiyun Gold Deposit

The area of the Baiyun gold deposit underwent complex tectonic movement and magmatic emplacement, forming a large area of emerging Neoproterozoic gneissic granitic rocks and Mesozoic large granite bases, rock masses, and dikes. Strata in the metallogenic belt have exhibited different degrees of regional metamorphism, among which ultrahigh-pressure metamorphic strata in the Dabie area are a major site for studies of intracontinental orogeny. The exposed metamorphic strata in the region mainly include a set of Neoproterozoic Hongan Rock Group schist and gneiss formations formed by intense metamorphism, such as plagio-amphibolite, biotite hornblende plagioclase gneiss, monzonitic gneiss, and granulite. The original rocks belong to the calc-alkaline basalt-dacite-rhyolitic volcanic series and siliceous rocks [13]. The large faults in the Tongbai-Dabie metallogenic belt are mainly the SEE-trending Xiaotian-Mozitan, Gueshan-Meishan, and Tongbai-Chengcheng faults and the NEE-trending Tanlu, Tuanfeng-Macheng, and Huanshui faults. A large number of secondary faults are also distributed in the NW, NE, and SW directions, forming the integral structural framework of the Tongbai-Dabie metallogenic belt. The Baiyun gold mining area is located east of the intersection of the NW-trending Xincheng Huangpi shear zone and the NNE-trending Huanshui fault [10–13]. Strong structural and magmatic activities gave rise to the unique structural characteristics of the Daleishan dome. The orebody occurs in a brittle fault zone with a steep dip in the NW and NW directions [12,13]. The ore-bearing surrounding rocks are Neoproterozoic Daleishan gneissic granodiorite, Mesozoic lamprophyre, and Huangmailing Formation of the Neoproterozoic Hongan Rock Group, and the rock types are relatively simple (Figure 2).

### 2.2. Deposit Geology

The Baiyun gold deposit is a medium-sized gold deposit (gold resources of 5.995 tons) located east of the Daleishan Dome within Dawu County. The exposed strata of the mine are mainly the Huangmailing and Qijiaoshan formations of the Neoproterozoic Hongan Rock Group, and the stratigraphic lithology is mainly dolomite albitite gneiss, quartz schist, albitite hornblende schist, albitite granulite, light granulite, manganese marble, graphite schist, and epidotite albitite hornblende schist [10–13]. The fault structure in the mining area mainly consists of four groups: NW-, NNW-, NE-, and NNE-trending faults [12,13]. The brittle-ductile shear zone near the EW- or NW-trending faults is the main ore-hosting structure, and it shows multi-stage activity characteristics. The orebodies mainly exist in late superimposed brittle faults, and rock breakage and alteration in the fault zone tend to contain high-grade ore. At the intersection of the NW and NE faults, mineralization is often more significant, and a small number of ore veins are produced in the NE faults, suggesting that the gold mineralization in this area is closely related to the formation time of the NE-trending fault (Figure 2). The magmatic rocks exposed in the area are mainly Neoproterozoic Daleishan intrusion rocks, which are batholith-shaped and have been transformed by deformation and metamorphism. The main lithology is a set of granitic and granodioric gneisses, and their petrochemical characteristics show that they belong to deep-source synfusion calc-alkaline granites. Lamprophyre intrusions (dikes) in different directions are widely distributed in the mining area, most of which are filled along the NE- and NW-trending faults. The scale of the dikes ranges from several meters to hundreds or

even thousands of meters, and the thickness is generally 0.50–10 m. Gold orebodies are produced in the contact area between the dike and surrounding rock.

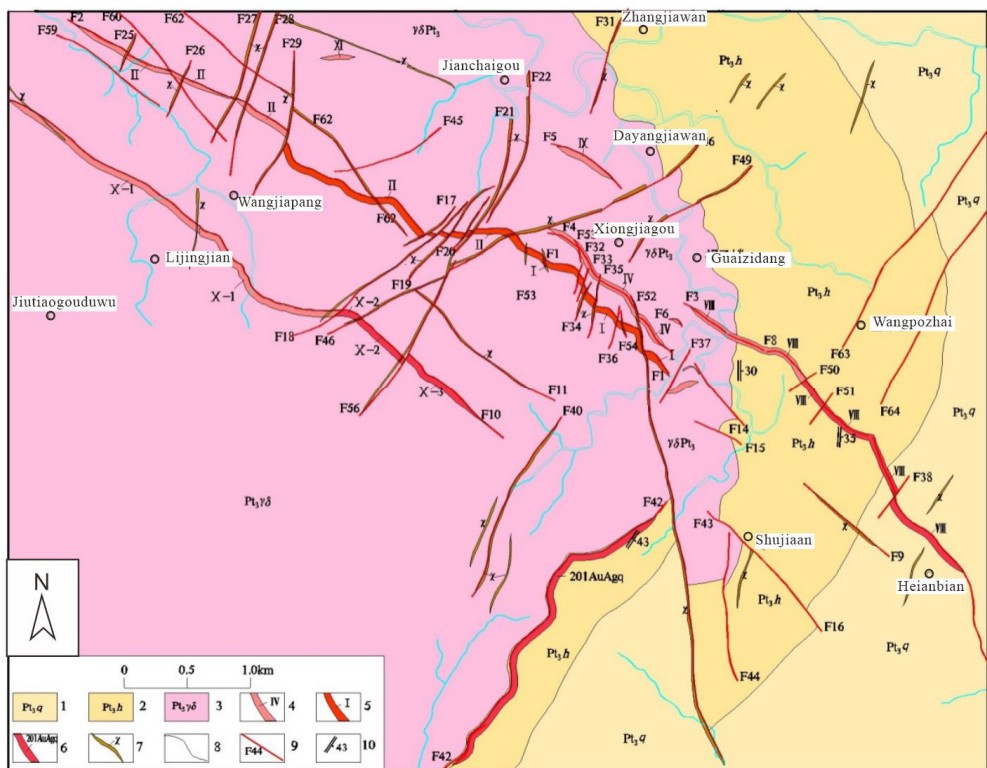

**Figure 2.** Geological map of the Baiyun gold deposit in NE Hubei Province (modified from [13]). 1—Qijiangshan Formation of Neoproterozoic Hongan group; 2—Huangmaling Formation of Neoproterozoic Hongan group; 3—Neoproterozoic granite diorite; 4—Au-bearing mineralized quartz veins and their No.; 5—Au-bearing quartz veins and their No.; 6—Au-Ag-bearing quartz veins and their No.; 7—Lamprophyre veins; 8—Geological boundary; 9—Faults and their number; 10—Foliation occurrence.

The entire ore-forming length is over 9 km, the width is approximately 2 km, and more than 10 veins are observed [10–13]. Five veins are found in the industrial orebody with a certain scale; the No. 201 vein is deposited in the NNE-trending interlayer fracture zone, and the No. I, No. II, No. VIII, and No. X veins are deposited in the quartz vein of the NW-trending fracture zone (Figures 2 and 3). The overall trend of the orebody is 290–325°, with a dip of 35–75° from west to east, and the dip angle changes from steep to slow and then to steep. The length of a single orebody is 1140–5300 m, with a thickness of 0.13–4.76 m, and the average grade is $1.27$–$9.79 \times 10^{-6}$. The average grade of associated silver is $10.91$–$235.85 \times 10^{-6}$. A single orebody is vein-shaped, plate-shaped, or plate-like, consists of multiple small orebodies, and is produced in a roughly parallel lateral form in space. The lateral direction is 260°–280°, and the lateral angle is 30°–40°. The mineralization is characterized by an uneven to extremely uneven distribution along the trend and the dip (i.e., from west to east), the gold content generally changes from low to high to low, the silver content gradually increases from low to high, and the Au/Ag ratio changes from 1:3.2 in vein II to 1:59 in vein VIII [10]. The change along the trend is greater than that along the dip. The ore types in the mining area are mainly quartz vein type and altered rock type. The ore minerals are mainly native gold, electrum, pyrite, galena, sphalerite, and chalcopyrite, whereas the gangue minerals are mainly quartz, sericite, Fe-dolomite, and calcite. The gold in the veins of the Baiyun gold deposit is mainly produced in the form of native gold and electrum, which are mostly symbiotic with pyrite, quartz, galena, and sphalerite. The mineral particle size is between 5 and 100 μm (i.e., micro-gold). The electrum particles are

liquid droplets and irregular strips filled in the cracks or gaps of pyrite and quartz particles, and a small amount is wrapped in sulfide minerals and quartz particles. The ore textures are mainly granular, disseminated, semi-automorphic–heteromorphic, contained, and granular metamorphosed. The ores are characterized by isolated granular, disseminated, zonal, veinlet, and cataclastic structures. The main wall-rock alterations in the ore district are silicification, potassium feldspathization, sulfidation, carbonatization, and illitization. Silicification is commonly seen on both sides of the gold-bearing quartz veins, with a width of 5–30 cm and locally up to about 1 m. Sulfidation is commonly seen on the side of the wall-rock near the gold-bearing quartz vein, and it generally shows a scattered, disseminated, and strip-shaped output. Moreover, areas closer to the gold-bearing quartz vein show stronger mineralization. Carbonatization is a type of alteration during late mineralization that mainly occurs in gelatinous-protocrystalline carbonate veins, which are produced in the fissures of the wall-rock, and gold-bearing quartz veins (Figure 4E).

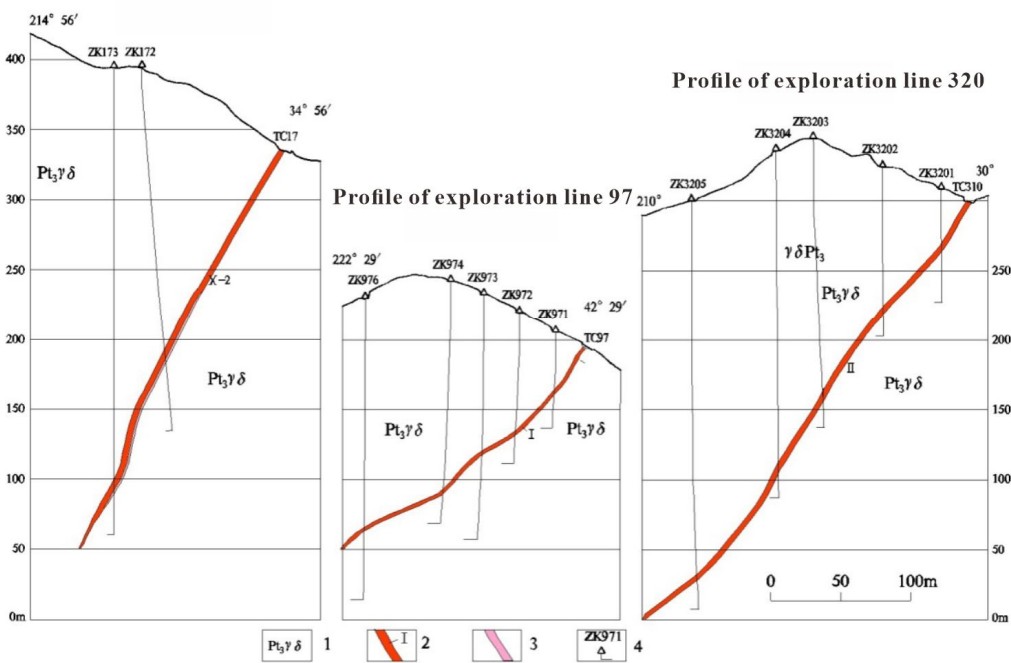

**Figure 3.** Profile of exploration lines 17, 97 and 320 in the Baiyun gold deposit (modified from [13]). 1—Neoproterozoic granite diorite; 2—Orebodies and their number; 3—Mineralized body and their number; 4—Drilling and their number.

Based on the mineral combination, formation sequence, ore texture and structure, relationship between wall-rock alteration and mineralization, and occurrence and interpenetration of ore veins, the Baiyun gold deposit can be divided into two metallogenic periods: hydrothermal metallogenic period and epigenetic metallogenic period. The hydrothermal mineralization period can be divided into three mineralization stages: early quartz-potassium feldspar stage, main quartz-native gold-electrum-porous pyrite-fine polymetallic sulfides mineralization stage (Figure 4A,B,D,E), and late quartz-rim-pyrite-coarse polymetallic sulfides-native silver-carbonate-illite mineralization stage (Figure 4A,C,D,F). During the supergene period, pyrite, chalcopyrite, and other sulfides were mainly oxidized to limonite and copper blue.

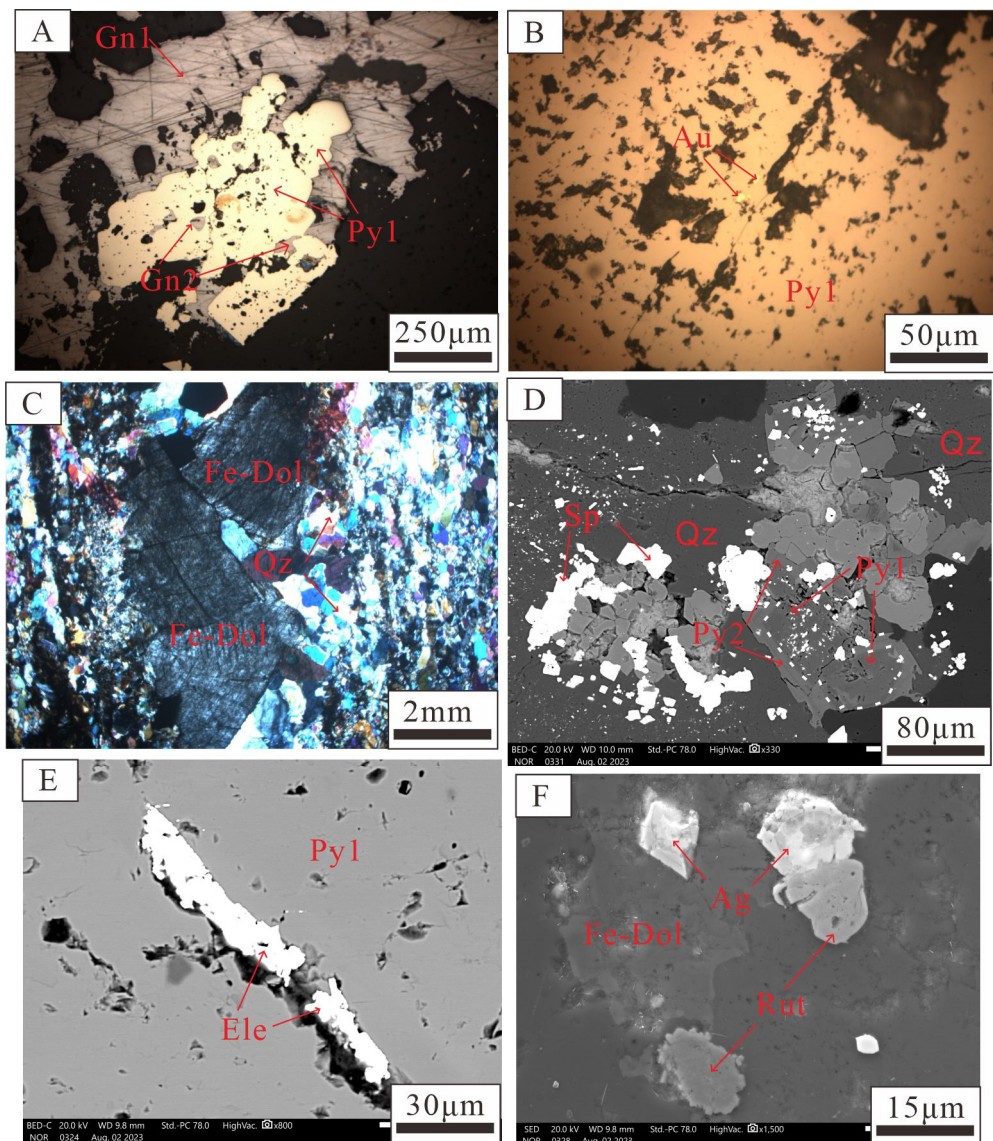

**Figure 4.** Morphological and textural features of different minerals in the Baiyun gold deposit including the (**A**) Porous Py1 contains Gn1 and symbiotic with Gn2 (RPPL image), (**B**) Porous Py1 contains natural gold particles (RPPL image), (**C**) The Fe-dolomite in the late ore-forming stage is filled along the fissures (RPPL image), (**D**) Porous Py1 contains fine-grained sphalerite and associated with coarse-grained sphalerite; Py2 overgrown by Py1. (BSE image), (**E**) Electrum is produced in Py fractures (BSE image), (**F**) Fine natural silver and rutile particles are produced in dolomite and quartz (BSE image). Abbreviations. Py: pyrite; Qz: quartz; Rut: rutile; Gn: galena; Sp: sphalerite; Au: natural gold; Ele: electrum; Ag: natural silver; Fe-Dol: Fe-dolomite. BSE: backscattered electron; and RPPL: reflected plane-polarized light.

## 3. Samples and Analytical Methods

### 3.1. Sample Preparation

Twelve ore samples were collected from the cross-section of an outcrop of the No.202 orebody (Figure 2), and 18 laser sections were prepared and polished to a thickness of approximately 100 μm for mineralographic and geochemical analyses. All sections were observed using optical microscopy and scanning electron microscopy (SEM) to identify the minerals, symbiotic relationships, and structural characteristics. Based on the generated data and comprehensive evaluation, coarse-grained sulfide minerals assigned to different ore stages were selected for in situ analyses using LA-ICP-MS. In situ analyses also involved

Fe and S isotopes of sulfides located adjacent to the associated sample used in the screening analyses, and the results are presented in Tables 1–3.

**Table 1.** Trace element data (ppm) for two types of pyrite samples from the Baiyun gold deposit obtained using LA-ICP-MS.

| Pyrite Type | Spot No. | Au | Co | Ni | Cu | Ti | Zn | As | Mo | Ag | Te | Bi | Pb |
|---|---|---|---|---|---|---|---|---|---|---|---|---|---|
| Py1 | BY-3-1-8 | 0.54 | 0.30 | 1.43 | 0.38 | 0.07 | 0.00 | 0.06 | 0.56 | 0.00 | 0.04 | 0.00 | 1.92 |
| | BY-5-1-1 | 0.55 | 0.36 | 1.57 | 1.58 | 0.61 | 0.00 | 0.07 | 0.82 | 0.02 | 0.09 | 0.01 | 3.42 |
| | BY-5-1-3 | 0.97 | 0.39 | 1.91 | 2.38 | 0.76 | 0.00 | 0.07 | 0.83 | 0.43 | 0.10 | 0.03 | 4.57 |
| | BY-5-1-4 | 1.12 | 0.68 | 2.27 | 4.26 | 0.97 | 0.00 | 0.08 | 1.43 | 1.49 | 0.19 | 0.04 | 7.71 |
| | BY-5-1-12 | 1.85 | 0.97 | 4.50 | 4.36 | 0.98 | 2.29 | 0.09 | 1.54 | 1.65 | 0.31 | 0.08 | 13.22 |
| | BY-2-1-1 | 2.30 | 2.07 | 7.33 | 4.50 | 1.03 | 3.25 | 0.14 | 3.15 | 1.78 | 0.45 | 0.14 | 15.99 |
| | BY-2-1-2 | 2.36 | 2.78 | 7.33 | 7.72 | 1.39 | 3.85 | 0.15 | 5.71 | 2.74 | 0.56 | 0.27 | 16.15 |
| | BY-2-1-3 | 2.46 | 4.80 | 8.49 | 9.92 | 2.04 | 3.90 | 0.16 | 7.38 | 4.22 | 0.63 | 0.30 | 19.93 |
| | BY-2-1-4 | 2.61 | 11.65 | 11.20 | 13.09 | 3.47 | 4.35 | 0.18 | 7.44 | 8.06 | 0.71 | 0.55 | 31.99 |
| | BY-2-1-5 | 2.94 | 21.90 | 24.09 | 41.57 | 26.60 | 5.38 | 0.19 | 19.57 | 12.04 | 0.81 | 1.21 | 41.70 |
| | BY-2-1-6 | 3.04 | 24.34 | 36.46 | 45.93 | 61.24 | 6.49 | 0.22 | 21.13 | 15.34 | 1.60 | 1.24 | 216.07 |
| | BY-2-1-7 | 3.60 | 51.45 | 41.48 | 46.30 | 221.48 | 7.47 | 0.36 | 23.79 | 15.71 | 1.82 | 1.64 | 326.25 |
| | BY-1-1-12 | 3.94 | 57.13 | 50.70 | 55.86 | 330.71 | 7.54 | 10.02 | 83.48 | 36.81 | 3.18 | 4.08 | 921.45 |
| | BY-1-1-13 | 8.29 | 59.67 | 53.14 | 86.77 | 366.47 | 7.95 | 32.40 | 90.06 | 57.53 | 3.36 | 12.16 | 6379.76 |
| | BY-1-1-14 | 33.56 | 68.46 | 86.66 | 490.07 | 1938.54 | 13.86 | 142.31 | 698.28 | 491.68 | 3.57 | 126.20 | 74353.59 |
| Py2 | BY-3-1-1 | 0.00 | 0.00 | 0.00 | 0.00 | 0.00 | 0.00 | 0.03 | 0.00 | 0.00 | 0.00 | 0.00 | 0.00 |
| | BY-3-1-2 | 0.49 | 0.00 | 0.00 | 0.00 | 0.00 | 0.00 | 0.05 | 0.00 | 0.15 | 0.00 | 0.00 | 0.00 |
| | BY-3-1-3 | 0.58 | 0.00 | 0.00 | 0.00 | 0.00 | 0.00 | 0.07 | 0.00 | 0.22 | 0.00 | 0.00 | 0.03 |
| | BY-3-1-4 | 0.65 | 0.51 | 0.00 | 0.01 | 0.00 | 0.00 | 0.09 | 0.00 | 0.30 | 0.00 | 0.00 | 0.03 |
| | BY-3-1-5 | 0.74 | 0.54 | 0.46 | 0.02 | 0.00 | 0.00 | 0.09 | 0.00 | 0.37 | 0.00 | 0.00 | 0.04 |
| | BY-3-1-6 | 0.83 | 0.68 | 0.52 | 0.13 | 0.00 | 0.00 | 0.10 | 0.01 | 0.40 | 0.00 | 0.00 | 0.07 |
| | BY-3-1-7 | 0.92 | 0.79 | 0.83 | 0.20 | 0.18 | 0.00 | 0.11 | 0.01 | 0.56 | 0.00 | 0.00 | 0.08 |
| | BY-5-1-2 | 0.96 | 0.88 | 0.90 | 0.30 | 0.40 | 0.05 | 0.13 | 0.02 | 0.60 | 0.01 | 0.00 | 0.09 |
| | BY-5-1-5 | 1.03 | 1.01 | 1.16 | 0.41 | 0.41 | 0.86 | 0.13 | 0.02 | 0.63 | 0.01 | 0.00 | 0.10 |
| | BY-5-1-6 | 1.35 | 1.58 | 1.19 | 0.43 | 0.44 | 1.30 | 0.13 | 0.02 | 0.69 | 0.01 | 0.00 | 0.13 |
| | BY-5-1-7 | 1.37 | 1.64 | 1.30 | 0.45 | 0.45 | 1.44 | 0.13 | 0.03 | 0.69 | 0.01 | 0.00 | 0.13 |
| | BY-5-1-8 | 1.55 | 1.66 | 1.30 | 0.51 | 0.55 | 1.49 | 0.14 | 0.04 | 0.70 | 0.01 | 0.00 | 0.23 |
| | BY-5-1-9 | 1.55 | 3.39 | 1.44 | 0.55 | 0.61 | 1.69 | 0.14 | 0.04 | 0.80 | 0.02 | 0.01 | 0.41 |
| | BY-5-1-10 | 1.56 | 3.64 | 1.45 | 0.60 | 0.62 | 2.12 | 0.16 | 0.06 | 0.91 | 0.02 | 0.01 | 0.58 |
| | BY-5-1-11 | 1.86 | 8.80 | 2.04 | 0.87 | 0.62 | 2.38 | 0.16 | 0.12 | 1.05 | 0.02 | 0.02 | 1.85 |
| | BY-2-1-8 | 1.86 | 11.27 | 2.62 | 0.88 | 0.69 | 2.58 | 0.16 | 0.24 | 1.15 | 0.02 | 0.02 | 2.36 |
| | BY-1-1-1 | 1.95 | 14.65 | 2.92 | 1.65 | 0.70 | 3.63 | 0.17 | 0.32 | 1.41 | 0.03 | 0.02 | 2.38 |
| | BY-1-1-2 | 2.04 | 18.62 | 3.01 | 1.67 | 0.71 | 4.57 | 0.17 | 0.39 | 1.60 | 0.03 | 0.02 | 2.41 |
| | BY-1-1-3 | 2.25 | 20.78 | 3.71 | 1.78 | 0.71 | 5.30 | 0.18 | 0.40 | 1.70 | 0.03 | 0.03 | 3.44 |
| | BY-1-1-4 | 2.25 | 22.33 | 6.67 | 2.03 | 0.73 | 5.39 | 0.18 | 0.69 | 3.50 | 0.05 | 0.03 | 5.15 |
| | BY-1-1-5 | 2.26 | 23.13 | 8.38 | 2.55 | 0.74 | 5.84 | 0.18 | 0.88 | 4.42 | 0.07 | 0.03 | 10.77 |
| | BY-1-1-6 | 2.29 | 36.99 | 9.28 | 3.33 | 0.77 | 7.85 | 0.18 | 0.94 | 5.78 | 0.08 | 0.04 | 15.79 |
| | BY-1-1-7 | 2.34 | 41.10 | 10.49 | 3.38 | 0.79 | 8.71 | 0.18 | 1.08 | 6.12 | 0.09 | 0.09 | 19.78 |
| | BY-1-1-8 | 2.55 | 56.15 | 12.16 | 4.30 | 0.90 | 8.87 | 0.19 | 3.27 | 6.43 | 0.11 | 0.13 | 21.82 |
| | BY-1-1-9 | 2.62 | 70.03 | 12.46 | 5.37 | 0.99 | 9.05 | 0.21 | 3.28 | 10.05 | 0.12 | 0.17 | 22.15 |
| | BY-1-1-10 | 2.68 | 78.42 | 17.72 | 5.43 | 1.04 | 9.99 | 0.21 | 5.43 | 11.35 | 0.12 | 0.18 | 24.44 |
| | BY-1-1-11 | 3.03 | 78.68 | 18.01 | 5.47 | 1.07 | 18.62 | 0.22 | 5.76 | 13.04 | 0.25 | 0.21 | 50.61 |
| | BY-1-1-15 | 3.84 | 160.83 | 29.51 | 8.70 | 1.14 | 36.80 | 0.24 | 6.90 | 16.59 | 0.64 | 0.46 | 67.65 |

Abbreviations: 0.00 = below detection limit.

**Table 2.** S isotope data (‰) for different types of pyrite and galena samples in the Baiyun gold deposit determined using LA-MC-ICP-MS.

| Types | Stages | Spot No. | $\delta^{34}S_{V-CDT}$ (‰) |
|---|---|---|---|
| Py1 | Main ore stage | BY-2-1-PY-1 | 0.79 |
| | | BY-2-1-PY-2 | 2.07 |
| | | BY-2-1-PY-3 | 0.41 |
| | | BY-2-1-PY-4 | −0.23 |
| | | BY-2-1-PY-5 | 2.19 |
| | | BY-2-1-PY-6 | 0.05 |
| | | BY-2-1-PY-7 | 1.81 |
| | | BY-2-1-PY-8 | 0.63 |
| | | BY-2-1-PY-9 | −0.16 |
| | | BY-1-1-PY-10 | 1.42 |
| | | BY-5-1-PY-11 | 1.16 |
| | | BY-5-1-PY-12 | 0.92 |
| | | BY-1-1-PY-4 | 0.15 |
| | | BY-1-1-PY-5 | 2.73 |
| | | BY-1-1-PY-6 | 2.79 |
| | | BY-1-1-PY-7 | 2.42 |
| | | BY-1-1-PY-8 | 1.06 |
| | | BY-1-1-PY-9 | 1.89 |
| | | BY-1-1-PY-10 | 1.57 |
| | | BY-1-1-PY-11 | 1.70 |
| | | BY-1-1-PY-12 | 3.04 |
| | | BY-1-1-PY-13 | 1.91 |
| | | BY-1-1-PY-14 | 0.94 |
| | | BY-1-1-PY-15 | 1.79 |
| | | BY-1-1-PY-16 | 1.26 |
| Py2 | Late ore stage | BY-1-1-PY-1 | 5.89 |
| | | BY-1-1-PY-2 | 4.80 |
| | | BY-1-1-PY-3 | 5.30 |
| | | BY-1-1-PY-4 | 6.09 |
| | | BY-1-1-PY-5 | 1.27 |
| | | BY-1-1-PY-6 | 3.51 |
| | | BY-1-1-PY-7 | 1.80 |
| | | BY-1-1-PY-8 | 5.45 |
| | | BY-1-1-PY-9 | 4.93 |
| Gn1 | Main ore stage | BY-1-1-GN-1 | 2.97 |
| | | BY-1-1-GN-2 | 3.31 |
| | | BY-1-1-GN-3 | 3.22 |
| | | BY-1-1-GN-4 | 3.08 |
| | | BY-1-1-GN-5 | 3.24 |
| | | BY-1-1-GN-6 | 3.34 |
| | | BY-1-1-GN-7 | 3.17 |
| | | BY-1-1-GN-8 | 3.47 |
| | | BY-1-1-GN-9 | 3.08 |
| | | BY-1-1-GN-10 | 3.16 |
| | | BY-1-1-GN-11 | 3.15 |
| | | BY-1-1-GN-12 | 3.30 |
| Gn2 | Main ore stage | BY-1-1-GN-13 | 3.16 |
| | | BY-1-1-GN-14 | 3.16 |
| | | BY-1-1-GN-15 | 3.30 |
| | | BY-1-1-GN-16 | 3.00 |

**Table 3.** Fe isotope data (‰) for two different types of pyrite samples in the Baiyun gold deposit determined using LA-MC-ICP-MS.

| Types | Stages | Spot No. | $\delta^{56}Fe_{IRMM014}$‰ |
|---|---|---|---|
| Py1 | Main ore stage | BY-1-1-1 | 0.23 |
| | | BY-1-1-2 | 0.18 |
| | | BY-1-1-3 | −0.05 |
| | | BY-1-1-4 | 0.38 |
| | | BY-5-1-1 | 0.45 |
| | | BY-5-1-2 | 0.43 |
| | | BY-5-1-3 | 0.05 |
| | | BY-5-1-4 | 0.31 |
| | | BY-5-1-5 | 0.37 |
| | | BY-5-1-6 | 0.34 |
| | | BY-2-1-1 | 0.33 |
| | | BY-2-1-2 | 0.47 |
| | | BY-2-1-3 | 0.33 |
| | | BY-2-1-4 | 0.36 |
| | | BY-2-1-5 | 0.21 |
| | | BY-2-1-6 | 0.33 |
| | | BY-2-1-7 | 0.36 |
| | | BY-3-1-1 | 0.50 |
| | | BY-3-1-2 | 0.26 |
| | | BY-3-1-3 | 0.11 |
| | | BY-3-1-4 | 0.21 |
| | | BY-1-1-1 | 0.23 |
| Py2 | Late ore stage | BY-1-1-5 | 0.42 |
| | | BY-1-1-6 | 0.82 |
| | | BY-1-1-7 | 0.60 |
| | | BY-1-1-8 | 0.41 |
| | | BY-1-1-9 | 0.27 |
| | | BY-1-1-10 | 0.27 |
| | | BY-1-1-11 | 0.50 |
| | | BY-1-1-12 | 0.49 |
| | | BY-1-1-13 | 0.45 |
| | | BY-1-1-14 | 0.47 |
| | | BY-5-1-7 | 0.61 |
| | | BY-5-1-8 | 0.59 |
| | | BY-5-1-9 | 0.57 |
| | | BY-5-1-10 | 0.52 |
| | | BY-5-1-11 | 0.59 |
| | | BY-5-1-12 | 0.45 |
| | | BY-2-1-8 | 0.66 |
| | | BY-2-1-9 | 0.65 |
| | | BY-3-1-5 | 0.41 |
| | | BY-3-1-6 | 0.36 |
| | | BY-3-1-7 | 0.25 |

*3.2. LA-ICP-MS Trace Element Analyses*

Analyses of the pyrite samples were conducted by Beijing Createch Texting Technology Co., Ltd. (Beijing, China) using a RESOlution 193 nm laser ablation instrument equipped with a COMPex Pro 102 ArF excimer laser and a MicroLas optical system. Details on the analytical instruments and procedures employed in the sample analysis have been reported previously [55]. An Analytik Jena PQMS Elite ICP-MS instrument was used to determine the signal intensities of different ions. He and Ar served as the ion carrier and makeup gas, respectively. They were mixed before their input into the T-connector, and the mixture was then released into the ICP. A signal smoothing component was installed in the LA system to produce a stable signal at a low laser repetition rate of 1 Hz [56]. The diametrical laser beams for the sulfide and calcite veins had laser beam sizes of 24 and

50 μm, respectively, and the laser repetition rate was 8 Hz. The compositional data for the sulfides were corrected using data from the USGS MASS-1 sulfide standard [57,58]. Each LA point was associated with 20 s and 45 s of background and sample–signal acquisition, respectively. Offline calculations of the concentrations were conducted using the ICP-MS Data Cal program [59,60].

### 3.3. In Situ S Isotope Analysis

The S isotope analyses of pyrite and galena were conducted using a Nu Plasma II MC-ICP-MS equipped with a Resolution-S155 excimer ArF LA system at Beijing Createch Texting Technology Co., Ltd. (Beijing, China). The 193 nm ArF excimer laser, which was homogenized via a set of beam delivery systems, was focused on the surface of the sample at a fluence of 2.5 J/cm$^2$. Each acquisition incorporated 30 s of background and 40 s of a spot with a diameter of 24 μm, with a repetition rate of 8 Hz. He (800 mL/min) was used as the carrier gas to efficiently purge aerosols from the ablation cell and was mixed with Ar (~0.8 L/min) in a T-connector before its entry into the ICP torch. The integration time of the Nu Plasma II system was 0.3 s. The Wenshan natural pyrite crystal (GBW07267) prepared by the Chinese Academy of Geological Sciences, Beijing, China [55] was utilized as a standard, and the reproducibility of the $\delta^{34}$S measurements was better than 0.6‰.

### 3.4. In Situ Fe Isotope Analysis

The LA-MC-ICP-MS in situ Fe isotope analysis was completed by Tianjin Kehui Test Technology Co., Ltd. (Tianjin, China). A Neptune Plus multi-receiver plasma mass spectrometer (Thermo Scientific, Waltham, MA, USA) with a RESOLution SE 193 nm solid laser was used. The detection environment required a temperature of 18–22 °C and a relative humidity of <65%. According to images of the sample after scanning, the appropriate area was selected and the sulfide was stripped using a laser stripping system. Stripping was performed by point stripping, with an adjustable stripping diameter of 10–150 μm, an energy density of 3 J/cm$^2$, and a frequency of 5 Hz. High-purity He was used as the carrier gas to blow out the aerosol produced by stripping, and the aerosol was sent to MC-ICP-MS for mass spectrometry. The Fe isotopes were simultaneously received by the static Faraday cup, and its structural parameters are listed in Table 1. The integration time was 0.131 s and 200 sets of data were collected, requiring approximately 27 s in total. Before the formal test, the instrument parameters were debugged with sulfide standard samples, including the plasma part (plasma tube position and carrier gas flow rate) and ion lens parameters. A highly sensitive injection cone combination ('X' skimmer cone + 'Jet' sample cone) was used to achieve a signal intensity for $^{56}$Fe of approximately 10 V in high-resolution mode. To minimize the matrix effect, sulfide standards similar to the sample matrix were used as standard samples (pyrite standard sample Tianyu-Py, pyrrhotite standard sample JC-Po, niobium pyrite standard sample JC-Pn, and chalcopyrite standard sample Tianyu-Ccp), and the standard sample–standard crossover method was used for quality discrimination correction [61–63].

## 4. Results

### 4.1. Occurrence and Texture of Pyrites

Pyrite is one of the main ore minerals in the Baiyun gold deposit. Two main pyrite types were identified based on chemical and textural analyses. The types and origins of the pyrites in the studied samples are depicted in Figure 4. The pyrite types, structures, compositions, and physical characteristics are described below.

#### 4.1.1. Py1

Py1 formed during the main stage of mineralization. SEM revealed an irregular shape and obvious dissolution pores and fissures. Some of the pores were filled with galena, sphalerite, and argon-gold formed during the same period, and the crystals were relatively coarse and often wrapped in pyrite formed in the late stage of mineralization.

### 4.1.2. Py2

Py2 formed in the late ore-forming stage, forming a typical girdling structure. This pyrite was irregular in shape and often grew around it during the main ore-forming stage. SEM revealed that Py2 was bright and smooth and that some pyrite particles were associated with fine sphalerite.

### 4.2. Occurrence and Texture of Galenas

According to the symbiosis with pyrite and mineral structure characteristics, galena can be divided into two types: fine galena coated with Py1 and disseminated coarse-grained galena grown around Py1.

### 4.2.1. Gn1

Gn1 is fine-grained and coated with Py1 in the main mineralization stage. It shows a subhedral-anhedral crystal structure, isolated distribution, and a particle size of less than approximately 100 μm.

### 4.2.2. Gn2

Gn2 particles are disseminated and closely symbiotic with Py1. The mineral particles are coarse and have a clear internal structure, and the pores inside the particles are developed. These pores are often filled with quartz, and they precipitate with Py1 at the same time in the main ore-forming stage.

### 4.3. Trace Elements of Pyrites Samples Determined Using LA-ICP-MS

The diameter of the beam spot of LA-ICP-MS used to test the in situ trace elements of pyrite was 24 μm. In total, 15 points were analyzed for Py1 and 29 points were tested for Py2. A correlation diagram of the data and mineralization elements is shown in Table 1 and Figure 5.

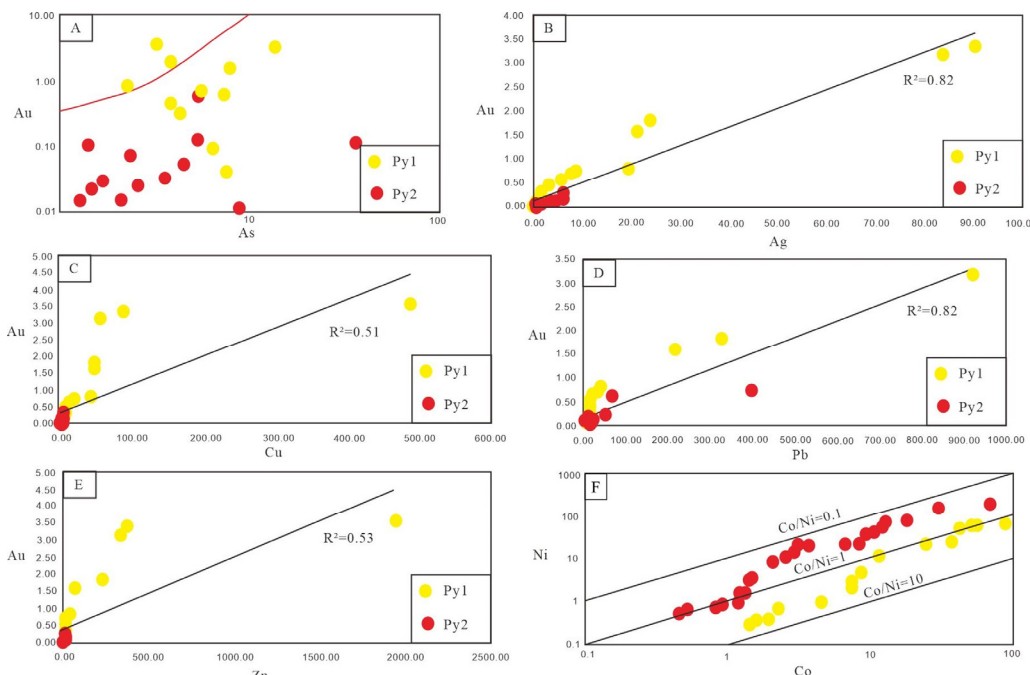

**Figure 5.** Elemental compositions of different types of pyrite in the Baiyun gold deposit and correlations including (**A**) As vs. Au, (**B**) Ag vs. Au, (**C**) Cu vs. Au, (**D**) Pb vs. Au, (**E**) Zn vs. Au, and (**F**) Co vs. Ni. The red line in Figure (**A**) means the maximum solubility of Au in pyrites. The black lines in the figures mean R2 correlation factors. The Au saturation line in (**A**) was obtained from Reich et al. [64]. The measuring unit of element contents: ppm.

The measured concentrations of Co, Ni, Cu, Ti, Pb, Zn, Ag, Au, As, Mo, Te, and Bi in the 44 pyrite samples are listed in Table 1. The trace elements of the main ore stage Py1 were Au (0.04–3.57 ppm), Co (0.30–68.46 ppm), Ni (1.43–86.66 ppm), Cu (0.38–490.07 ppm), Ti (0.54–33.56 ppm), Zn (0.07–1938.54 ppm), As (0.00–13.86 ppm), Mo (0.06–142.31 ppm), Ag (0.56–698.28 ppm), Te (0.00–491.68 ppm), Bi (0.00–126.20 ppm), and Pb (1.92–74353.59 ppm) (Table 1). The trace elements of the main ore stage Py2 were Au (0.00–0.73 ppm), Co (0.00–199.52 ppm), Ni (0.00–67.30 ppm), Cu (0.00–16.94 ppm), Ti (0.00–11.93 ppm), Zn (0.00–1.19 ppm), As (0.00–66.56 ppm), Mo (0.03–0.37 ppm), Ag (0.00–8.49 ppm), Te (0.00–21.20 ppm), Bi (0.00–0.82 ppm), and Pb (0.00–399.20 ppm) (Table 1). Compared with those for Py2 in the late ore-forming stage, Py1 in the main ore-forming stage had higher contents of Ni, Cu, Ti, Pb, Zn, Ag, Au, Mo, Te, and Bi but lower contents of Co and As and a lower Co/Ni ratio.

### 4.4. Sulfur Isotope Ratios of Pyrite and Galena Samples

In situ S isotope tests were conducted for two types of pyrite and galena samples using LA-MC-ICP-MS. The results are shown in Table 2 and Figure 6. The Py1, Py2, Gn1, and Gn2 samples yielded $\delta^{34}S$ values varying from −0.23‰ to 3.04‰ (mean = 1.37‰), 1.27‰ to 6.09‰ (mean = 4.34‰), 2.97‰ to 3.47‰ (mean = 3.21‰), and 3.00‰ to 3.30‰ (mean = 3.15‰), respectively. These results show that the in-situ S isotope data for the two galena types were concentrated, and no obvious fractionation was observed. Py2 in the late ore-forming stage has a larger sulfur isotope value than that for Py1 in the main ore-forming stage.

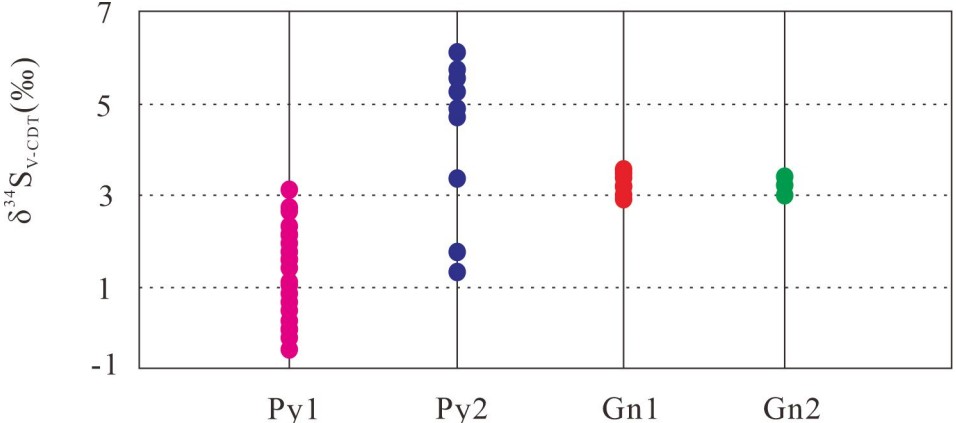

**Figure 6.** Plot displaying sulfur isotope ratios (‰) of different types of pyrite and galena from the Baiyun gold deposit.

### 4.5. Iron Isotope Ratios of Pyrite Samples

In situ Fe isotope tests were conducted for the two types of pyrite using LA-MC-ICP-MS. The results are shown in Table 3 and Figure 7. Py1 and Py2 samples yielded $\delta^{56}Fe$ values varying from −0.05‰ to 0.50‰ (mean = 0.29‰) and from 0.25‰ to 0.82‰ (mean = 0.49‰), respectively. The in situ Fe isotope values of Py1 and Py2 were both within the range of Fe isotopes of magmatic-hydrothermal pyrite calculated previously. However, compared with that for Py1 in the main ore-forming stage, Py2 in the late ore-forming stage had a relatively high Fe isotope value.

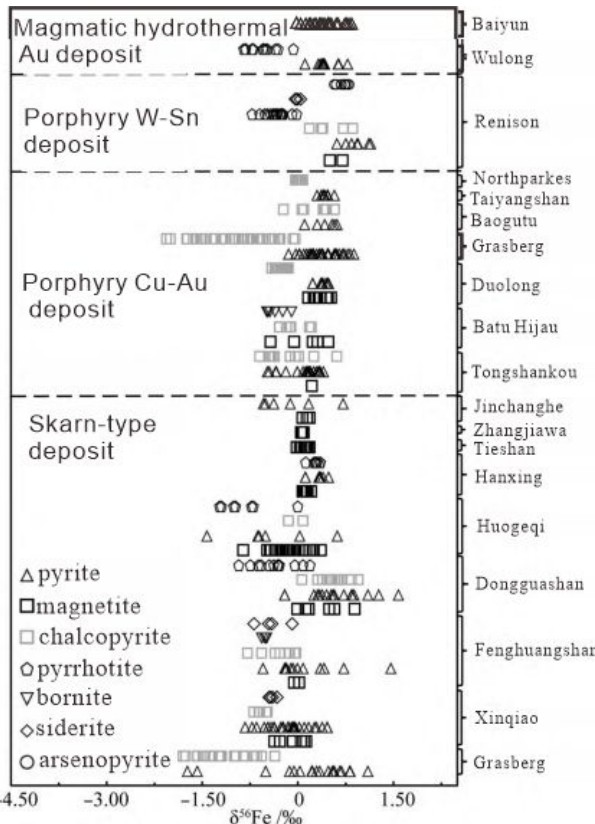

**Figure 7.** Iron isotope composition of iron-bearing minerals in different magmatic-hydrothermal deposits (modified from [65]).

## 5. Discussion

### 5.1. Genesis of Pyrite and Mechanism of Gold Precipitation

Pyrite is common in hydrothermal mineralization systems. During the formation of ore deposits, other elements can be precipitated simultaneously [66,67], which can effectively control the distribution of a series of economic trace elements [68,69], such as Au, Ag, As, Pb, Zn, and Cu, which can enter the pyrite lattice (solid solution, micron/nano-level mineral inclusions) [64,70–72]. Thus, research on pyrite can provide a better understanding of the characteristics and zoning of metals and can clarify the mechanism of superimposed mineralization events driven by physical and chemical processes (hydrothermal alteration, metamorphism, deformation, etc.) and changes in the composition and distribution of trace elements in ore-forming fluids [70,73–75]. Thus, the composition and microstructure of pyrite are useful complements to other geochemical information and provide insight into the evolutionary process of hydrothermal systems [57,74,76–78].

The Co and Ni contents in pyrite can reflect changes in the fluid temperature; higher temperatures are favorable for the replacement of Fe with Co and Ni in the pyrite lattice [67,70–82]. In addition, the Te content in pyrite shows a negative correlation with the $fO_2$ of fluids [83,84]. The Te content in stage 2 (average value $3.88 \times 10^{-6}$) was significantly lower than that in stage 1 (average value $44.30 \times 10^{-6}$), indicating that the $fO_2$ of fluids in stage 2 was higher than that in stage 1. Furthermore, the trace element content of pyrite can reflect the physical and chemical processes occurring in the fluids. For example, pyrite formed during the rapid boiling process has high Cu, Pb, Ag, and Au concentrations and low Co and Ni concentrations (Table 1) [67]. In addition, in terms of morphology, the porous structure and abundant mineral micro-inclusions (mainly galena and sphalerite) in pyrite (Py1) particles may indicate fluid boiling [68,85]. Pyrite (Py2) formed during the fluid mixing process was mostly the product of slow crystallization that occurred under

relatively stable physical and chemical conditions, and it is characterized by a well-formed proliferation zone that generally did not show bubbles and mineral inclusions.

The ratios of different elements can also be used to determine the source of pyrite and infer the metallogenic evolution and genesis of the deposit. The Co/Ni ratio of pyrite is an important marker for ensuring changes in the metallogenic environment, particularly as a source of hydrothermal gold deposits [69,86,87]. In degassing deposits related to basic intrusions, the Co and Ni contents in pyrite are relatively high. The Co/Ni ratio of the magmatic source is generally higher than that of pyrite formed in a sedimentary environment [69,87–90], with magmatic-hydrothermal pyrite exhibiting a high Co/Ni ratio (>1) but sedimentary pyrite exhibiting a low Co/Ni ratio (<1) [69,89]. The Co/Ni ratios of pyrite in the two metallogenic stages obtained in this study were mostly greater than 1, indicating that the source of the pyrites is magmatic-hydrothermal fluids with potential metasomatism with the old strata.

The Co, Ni, Cu, Ti, Pb, Zn, Ag, Au, As, Mo, Te, and Bi contents in the pyrites in the main metallogenic stage (Py1) were higher than those in the late metallogenic stage. Pyrites in this stage had an obvious porous structure, and a large number of metal sulfides (such as chalcopyrite, galena, and sphalerite), native gold, and electrum inclusions were found, indicating that fluid boiling may have occurred in this stage and a large number of minerals precipitated. The high Te content, Co/Ni ratio, and $\delta^{34}$S value showed that the mineralizing substances were mainly derived from magma, which may react with the surrounding rocks. Pyrites in the late metallogenic stage (Py2) had relatively low mineralized element contents and ring-like structures. There were fewer sulfide inclusions, such as sphalerite and galena, in the pyrites in this stage, indicating that the physical and chemical conditions of the fluids in this stage are relatively stable. The relatively higher Te content, Co/Ni ratio, and $\delta^{34}$S value show that the mineralizing substances also mainly come from magmatic-hydrothermal fluids. In addition, in the two stages, the Au content was positively correlated with the Cu, Pb, Zn, Ag, and As contents, and the gold content in some pyrites was above the Au-As solid solution dissolution line, indicating that gold in the deposit mainly occurred in the form of granular gold (natural gold and electrum), although structural gold or solid solution gold was also observed.

### 5.2. Sources of Sulfur and Iron

The $\delta^{34}$S value of pyrite is often used to track the source of minerals and to discuss the mineralization process [91–93]. Typical $\delta^{34}$S values for magma sulfur are −5‰ to +5‰, for sedimentary sulfur are <0‰, and for marine sulfur are +20‰ [94–96]. The S isotopic composition of hydrothermal sulfides is greatly affected by $fO_2$, pH, temperature, and primary fluid composition but is less affected by pressure [91,95]. Increasing data have revealed the Fe isotopic composition and variation of various geological reservoirs, magma minerals, rocks, and hydrothermal fluids/minerals [97–99]. Theoretical calculations and experimental research have also yielded some mineral–melt, mineral–fluid, and mineral–mineral Fe isotopic equilibrium fractionation coefficients [100–105]. These results provide a basis for the development of geological applications of Fe isotopes. Currently, Fe isotope data are primarily used in analyses of biological processes, environmental changes, and magma evolution, among other areas of research [106–111]. The application of Fe isotopes in mineral deposits mainly focuses on stripe iron formation; submarine hydrothermal mineralization, magmatic mineralization, magmatic-hydrothermal mineralization, and other hydrothermal mineralization systems; and the supergene oxidation of mineral deposits [112–122]. These studies show that Fe isotopes have great potential for tracing the source of mineralization materials and constraining the genesis of mineral deposits. The Fe isotopic composition of the main Fe-containing minerals (pyrite, magnetite, pyrrhotite, chalcopyrite, etc.) has been evaluated in magmatic-hydrothermal deposits.

The $\delta^{56}$Fe values of Fe-containing minerals in magmatic-hydrothermal deposits vary greatly (−2.07‰ to +1.58‰) [97–99]. The $\delta^{34}$S of pyrite in the main stage of mineralization was −0.23‰ to +3.04‰, with an average value of +1.37‰, which was within the range

for magmatic sulfur. The $\delta^{34}S$ of pyrite in the late stage of mineralization was +1.27‰ to +6.09‰, with an average value of +4.34‰, which was generally consistent with the range for magmatic sulfur, and some values exceeded the range for magmatic sulfur slightly, indicating that the sulfur in this stage mainly comes from magma, and some stratigraphic sulfur is added, indicating that fluid mixing occurs in the late stage of mineralization. The $\delta^{56}Fe$ of pyrite in the main metallogenic stage was −0.05‰ to +0.50‰, with an average value of +0.29‰, and the $\delta^{56}Fe$ of pyrite in the late metallogenic stage was +0.25‰ to +0.82‰, with an average value of +0.49‰, which was within the range for magmatic iron, indicating that the iron in this stage was mainly derived from magma.

*5.3. Genesis of the Deposit*

Systematic studies on the mineralogy, mineragraphy, in situ trace elements, and isotopes of polymetallic sulfides showed that the main ore minerals of the Baiyun gold deposit are native gold, electrum, pyrite, sphalerite, and galena (Figure 8). Based on the morphology, Co/Ni ratio, and geochemical characteristics of the pyrite, it is believed that the pyrite in the ores is mainly divided into two stages: the main metallogenic stage Py1 and the late metallogenic stage Py2. Py1 had a highly porous structure and contained a large amount of native gold, electrum, and metal sulfides (mainly sphalerite and galena), indicating that the pyrite in this stage was formed under the action of fluid boiling, which is the main factor controlling gold precipitation. Py2 was formed under a relatively stable fluid condition, and pyrite in this stage develops a typical annular structure with high brightness and a smooth surface under a SEM. This is the late metallogenic stage, and the contents of Co, Ni, Cu, Ti, Pb, Zn, Ag, Au, As, Mo, Te, Bi, and other mineralization elements were lower than those of Py1. The in situ S and Fe isotope tests of the two-stage pyrite showed that the ore-forming material was derived from the magma fluid. Previous researchers have used quartz veins of the Baiyun gold deposit as the research object, revealing that the metallogenic temperature of the Baiyun gold deposit was 170–350 °C and thus it belonged to the medium-low temperature hydrothermal gold deposit [10,11]. H-O isotope mapping revealed that most of the points were distributed in the range of magma water and a small part of the transition area between magma water and meteoric water, indicating that the ore-forming fluid mainly came from magma hydrothermal fluid, and meteoric water participated in the mineralization process [10,11]. Previous researchers have applied the quartz inclusion Rb-Sr dating method to the Baiyun gold deposit to confirm that the metallogenic age is 131.0 ± 10.0 Ma [123]. As mentioned previously, lamprophyre is an important ore-bearing rock in mining areas. According to the phenomenon that the quartz veins fill the cracks of lamprophyre, U-Pb dating of zircon in lamprophyre was carried out, which limited the metallogenic age of the Baiyun gold deposit to 126.8 ± 2.0 Ma [124]. This age is basically consistent with that obtained using quartz inclusion dating, indicating that the metallogenic age is the Early Cretaceous, consistent with the Yanshan period Pacific Plate subduction and the regional large area chalk granitic magma intrusion time.

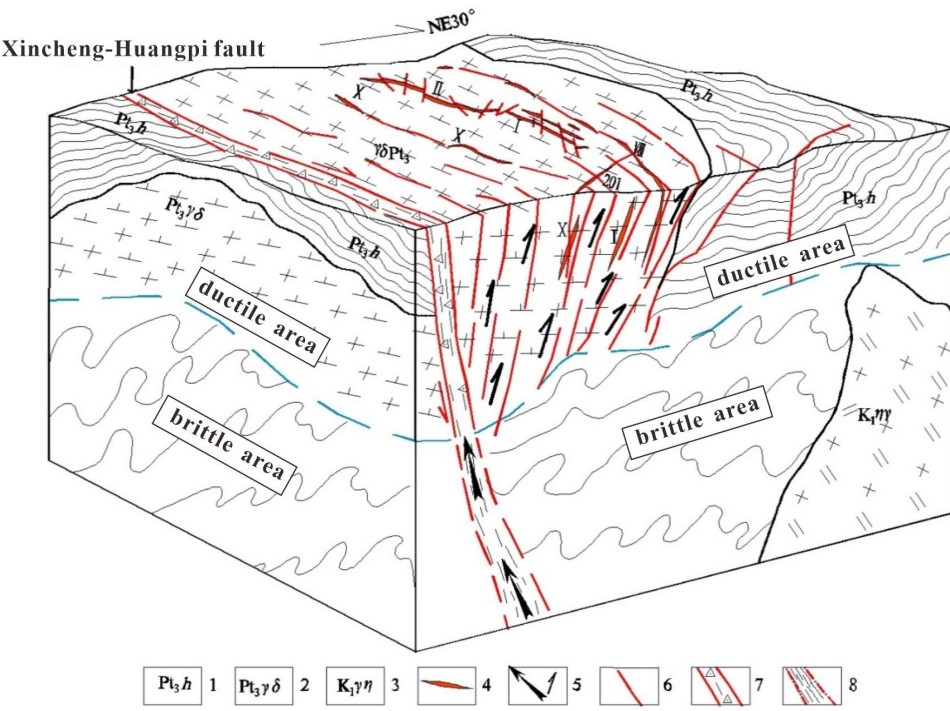

**Figure 8.** The metallogenic model of the Baiyun gold deposit (modified from [13]). 1—Neoproterozoic Huangmalingshan Formation; 2—Neoproterozoic Daleishan intrusive rock; 3—Cretaceous intrusive rock; 4—gold orebodies; 5—direction of ore-bearing hydrothermal migration; 6—brittle fault; 7—shallow brittle fracture zone; 8—deep ductile shear zone.

## 6. Conclusions

Two types of pyrite and galena were identified in the Baiyun gold deposit. Analyses of mineral morphology and mineral symbiosis suggest that fluid boiling occurred during the main stage of mineralization in the Baiyun gold deposit and was the main factor controlling gold precipitation. The gold content in some pyrites exceeded the Au-As solid solution dissolution line, indicating that gold in the deposit mainly exists in the form of grain gold (natural gold and electrum), with some in the form of structural or solid solution gold. Au was strongly positively correlated with Cu, Pb, Zn, Ag, As, Te, and other elements, indicating that these elements can be used as indicators of gold mineralization. The two types of pyrite had high Te contents and Co/Ni ratios. The in situ S and Fe isotopes of pyrite and the in situ S isotope of galena were all within the range of values for magma, indicating that the mineralization material originated from magmatic-hydrothermal processes. We concluded that the Baiyun gold deposit in Dawu County, Hubei Province is a magmatic-hydrothermal deposit formed in the Early Cretaceous.

**Author Contributions:** Author Contributions: Conceptualization, Investigation and Methodology, Data curation, Writing—original draft, Writing—review and editing, W.S.; Funding acquisition, W.S., J.L. and Y.Z.; Investigation and Methodology, X.L., T.L., J.Z., S.F. and S.C.; Project administration, W.S., J.L., Y.X., R.Z., J.Y., X.Z., Z.Y. and J.F.; Experimental data consolidation, W.S.; Writing—original draft, W.S.; and Writing—review and editing, W.S., J.L. and S.C.; All authors have read and agreed to the published version of the manuscript.

**Funding:** This study was jointly funded by Natural Science Foundation project of Hubei Province (2023AFD230), Science and technology project of Hubei Geological Bureau (KJ2024-28), and the Talent Team Program of Science and Technology Foundation of Guizhou Province (Qian Ke He Ping Tai Ren Cai CXTD [2021]007).

**Data Availability Statement:** Data are contained within the article.

**Acknowledgments:** We are grateful to E. Hongfang Chen for assistance during the BSE, SEM, and EDS analyses and E. Dewei Kong for the support provided when conducting the trace element and isotopic analyses of the sulfides. We would like to thank the anonymous reviewers for their constructive comments.

**Conflicts of Interest:** The authors declare no conflict of interest.

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
