# Peer review of "Genesis of the Baiyun Gold Deposit in Northeast Hubei Province, China: Insights from In Situ Trace Elements and S-Fe Isotopes of Sulfide"

_minerals, doi:10.3390/min14050517_

Round 1

Reviewer 1 Report

Comments and Suggestions for Authors

Dear Editor, Journal of Minerals,

Manuscript Number: minerals-2956183

Genesis of the Baiyun Gold Deposit in Northeast Hubei Province, China: Insights from In Situ Trace Elements and S-Fe Isotopes of Pyrite

The manuscript reports geology, trace Elements and S-Fe Iso-topes of pyrite and galena in the Baiyun Gold Deposit, China. Hence, the manuscript and its content have some contributions in understanding the geology and genesis of epithermal gold deposits. However, the presentation of the data and discussions in the manuscript requires substantial improvement. The authors do not have high-quality data in hand, except for some LA-ICP-MS analysis and scarce S-Fe isotopes on pyrite and galena, which are not systematic data set to cover all stages of alteration and mineralization, hence their description is not as good as expected. The major problem of the paper is that it is not internally coherent and the interpretations in some parts are questionable and just repetitions of previously published papers, which relatively makes the discussions and conclusions unreliable. In terms of terminology, there are many typos and grammatical errors that need to be fixed. I will not check more but the authors should check inconsistency thoroughly. In general, the manuscript needs to be clearer in its presentation based on new and comprehensive analytical data. Some problems need to be solved hence the following “Main Comments and Suggestions” and the attached “annotated pdf file” (for both manuscript and Figures) were provided in detail to improve the manuscript. It needs a significant improvement before it can be accepted for publication in this journal. Thus, considering all these, I recommend a significant improvement and Major Revision on the MS.

Main Comments and Suggestions:

1)    Please provide some good evidence, including field and microphotos to be relevant candidate for your mineralogical and textural declarations.

2)    It is highly recommended that authors provide a representative paragenetic sequence for the events and mineral associations, then provide their microphotographs IN ORDER. In that case the readers can feasibly follow the geological, alteration and mineralization stages in this deposit. A paragenetic sequence for this deposit is needed to identify the type of different gangue and ore minerals and related alteration stages obviously. Need to back up the mineral paragenesis with evidence.

3)    Why do the authors only analyze pyrite and galena? What is the reason that you only chose these two minerals for measurement of S isotopes.

4)   Please provide some photos for the spot of laser and also the textural relationship between Py I and Py II for such LA-ICP-MS.

5)    The authors never talked about the relationship between the two types of galena?! so such classification and running S isotopes for these two unknown phases are meaningless. Actually, I have no idea what its significance. Hence, their S isotopes data do NOT show something different which seems to be the same!

6)    Only by some trace elements contents in pyrite you can’t discuss about boiling condition in this ore system, in fact other strong evidence (like fluid inclusion data) need to be involved to fulfill such issue. However, other factors can control such trace elements distribution in sulfides like pH and Eh which mainly act individually.

7)    It is highly recommended that instead of talking about such simple and routine writing (like the physical and chemical conditions of the fluids in this stage are relatively stable), based on their trace elements contents and isotopic data the authors should provide some new information about Eh-pH conditions during the evolutionary trend of ore fluids in this deposit that may have affected gold precipitation. Basically, these are the interesting things that an international reader is looking for.

8)    ACTUALLY, THIS IS NOT A SYSTEMATIC STUDY! The authors only chose some random samples from the main and late stages of this deposit. not form pre ore, main ore, post ore stages! Since you don’t have a clear and complete mineralogy stagy and paragenetic sequence that clearly show the different stages, judgments about these phases are meaningless.

9)    Keep the order between the photos and rearrange the text! please see the attached “annotated pdf file”.

Comments on the Quality of English Language

The English structure needs moderate revision. 

Reviewer 2 Report

Comments and Suggestions for Authors

In this manuscript, in situ S and Fe isotopes of sulfide are used to study the genesis of the Baiyun gold deposit in Northeast Hubei Province, China. This manuscript has sufficient evidence, reasonable discussion and credible conclusions. It is suggested that the following contents be modified and accepted for publication after minor revisions.

(1) There is a large amount of Chinglish in this manuscript, please revise and polish the English of the whole text.

(2) This manuscript contains too much text and is too long. Please delete unnecessary text. For example, there are many repeated statements in the Abstract.

(3) Please significantly cut the content of 2. Geological background. Please keep only those statements that are closely related to this study.

(4) The legend in Figure 2 has translation errors, such as 10-Flacial occurrence.

(5) There is Chinese in Figure 3, and the legend is incomplete.

(6) The element content of the axis in Figure 5 has no unit.

(7) Figure 7 is too vague.

(8) Please draw a model diagram to show the important results of this study, such as the metallogenic model diagram.

Comments on the Quality of English Language

English needs to be improved.
